# OpenReview forum: "RL-Guard: Rescuing LLM Agents from Pitfalls"
_ICLR.cc/2026/Conference — ICLR 2026 Conference Withdrawn Submission_

### Official Review · Reviewer_wDf2 · 2025-10-21

**Soundness:** 1
**Presentation:** 2
**Contribution:** 2
**Rating:** 2
**Confidence:** 4

**Summary:**

This paper presents a method for safeguarding LLM agents against many types of potentially harmful actions. The paper proposes a RL-training methodology to train an actor, critic, and reward function. The actor and critic models will then be used as a safeguarding wrapper at inference time. In order to train these models with RL, the authors contribute a novel massive dataset of safe-unsafe trajectory pairs with individual actions labeled as safe or unsafe. The proposed methodology is evaluated on existing agent safety benchmarks and shows some Pareto increases in the safety-utility trade-off.

**Strengths:**

* The results are strong on the ToolEmu benchmark. In most cases, RL-Guard provides Pareto improvements over all baselines.
* The use of domain-specific risks as part of the safety reward function is very interesting (though does not necessarily promote generalization.
* The reward function learned using RL-Guard seems to be very good (evidence from Figure 6).
* The contribution of a massive (orders of magnitude larger than previous data sets) training set for training safeguards is an important contribution.

**Weaknesses:**

* The paper lacks soundness between its claims in the introduction and the experimental results provided. For example, the authors claim on line 082/083 (between lines) that RL-Guard's design enables generalization to unseen risks, yet there is no direct evidence provided by the authors that this generalization occurs. In fact, the opposite seems to be true: RL-Guard does not seem to generalize to AgentHarm very well at all, but it does well on ToolEmu which is very similar to the training data. There is no evidence that this method generalizes to unseen risks.
* There are many inaccuracies, ambiguities, and errors throughout that do not inspire confidence in this submission. For example, on lines 422-424, the authors state that in Table 4, GPT-4o + RL-Guard has the lowest harmful request score and highest harmful refusal rate on AgentHarm. Looking at Table 4, we can see that this is clearly not the case. Instead, gpt-oss-20B + RL-Guard is best on AgentHarm. Another example of ambiguities is the lack of a parameter size on "LLaMa-3.1-Instruct". Which model is this? Why is it spelled differently throughout the paper? Furthermore, what is going on with the rows in Table 7? Why is 4 candidates better than 5 and the rows ordered in that way?
* Only a single model is used to implement the proposed safeguard, and it is a non-standard open-weight model (OPT-6.7B).
* A lot of space in the paper is dedicated to describing the RL training (i.e. PPO), but this is fairly standard knowledge and is somewhat irrelevant to the paper at large. This should probably be moved to the appendix to make space for more analysis, experimental results, or even more in-depth exposition on the rest of the proposed methodology. In the submission form it is a little confusing how the trained safeguard is used at inference time. Maybe more space could be dedicated to that.
* It's unclear why we have a trained actor, critic, and reward model. Alternatively, we could just prompt a large model to act as any one of these components without the need for RL. We could also use a prompt optimization method (e.g. GEPA) guided by the RL-Guard benchmark. There should either be justification for why these obvious baselines are not explored or the results should be added to the paper.
* The paper writing could use some work. In its current state, it's unclear if the proposed method is training the agent or a safeguard. This was somewhat confusing and took multiple reads to understand. (Also several typos, see below for examples)
* This proposed method seems to be intended more for safety issues where the agent is making mistakes that cause harm and not guarding against misuse by malicious users. Maybe the paper's utility should be recast towards the more ToolEmu/R-Judge style of work.
* As with many other safeguarding methods, it seems like the proposed approach degrades utility quite significantly on OOD benign queries (looking at Table 4).

### Minor:
* Line 214: "illustrate" -> "illustrates"
* Line 269: "feedbacks" -> "feedback"
* Line 322: "Llama-3.1", but Line 333 (and the rest of the paper) "LLaMa-3.1-Instruct". (this is mostly an underspecification issue)
* RL-Guard Benchmark: "Benchmark" might not be the right word here. This is just a training set?
* Relevant related work: Yueh-Han Chen, Nitish Joshi, Yulin Chen, Maksym Andriushchenko, Rico Angell, and He He. "Monitoring Decomposition Attacks in LLMs with Lightweight Sequential Monitors." arXiv 2025.

**Questions:**

* What's the difference between the critic model and the reward model? Why do you need both? It seems like both are accomplishing a very similar goal. Are you just trying to fit in PPO? Or is the difference between them that you have harm-specific rewards, but do not have a harm-specific critic?
* Why use token-level rewards? Wouldn't action-level rewards make much more sense? The agent will take multiple actions (represented by multiple tokens) and we can just label each of these high-level actions as safe or unsafe instead of specific tokens. This actually seems like it would yield worse generalization.
* Why not just use self-reflection of the agent model? After the agent proposes an action just ask it if the proposed action could lead to any potential harms. Or just have a separate learned prompt for each type of harm?
* Why use the same single small open-weight model as the backbone for the actor, critic, and reward model?
* On lines 312-313, the authors state that the model first generates a *safe* trajectory. How do you generate a safe trajectory?
* How easily could new forms of harm be incorporated? Do we need to label more training data?

---

### Official Review · Reviewer_kr7G · 2025-11-01

**Soundness:** 2
**Presentation:** 3
**Contribution:** 2
**Rating:** 2
**Confidence:** 4

**Summary:**

The paper proposes RL-Guard, a framework that aims to detect and correct unsafe agent behavior. The system equips an agent with two additional components: a critic that scores agent action and trigger action regeneration when the score indicates unsafe condition, an actor that generates safe score and chooses the best answer after new candidates are regenerated. In order to train the critic and actor, the paper introduces RL training. For RL training, the paper proposes a large, synthetic dataset with safe and unsafe trajectory pairs. The generated trajectory is decomposed stepwise and then annotated by its risk. The framework then utilizes such a dataset to train the models. The paper compares its proposed method with other methods on their dataset and ToolEmu, Results show that the proposed framework boosts safety performance while causing a bit harm to the utility. The paper also adds a simple experiment on AgentHarm to demonstrate their method.

**Strengths:**

- The paper is well written and easy to follow.
- The paper targets agent safety, which is an important and emerging problem.
- The attempt to use RL to train models for safeguarding agent is interesting

**Weaknesses:**

- The design that the system needs both actor and critic seems weird, because the two models seem to do the same thing: rank the output of the agent model. Different from the usual 'actor-critic' framework we see in RL, the 'actor' is also scoring the output of the agent. This makes the fundamental design of the system weird: if the two module has the same function, one can simply use generation-evaluation loop:  the critic can simply evaluate the regenerated agent output and select the best one.

- The experiment is not sufficient. Firstly, evaluating directly on RL-Guard benchmark is flawed: I believe this is testing on training data(correct me if I am wrong here), which enables data leakage and gives your framework unfair advantage over other methods. After excluding this dataset, there is ONLY one dataset that is used to compare different methods, though Table1 listed 7 different datasets. Moreover, there is a Table 6 (that is not mentioned anywhere else in the paper) that compares computational cost ONLY with one baseline AgentMonitor, on a dataset AgentHarm that is not used for comparing model performance. This reporting strategy sounds weird. If AgentHarm has been tested for AgentMonitor, then its performance should also be included in the results (Table 4). Moreover, for a more comprehensive evaluation on computational cost, more frameworks should be added. The author may consider adding at least one (or two, if time permits) dataset for fairer evaluation.

**Questions:**

- See Weaknesses

- The labeling strategy of the instances in RL-Guard may require additional explanation. It now uses a labeling from 1 - 3. I am wondering why the 3-level design is chosen.

- The regeneration currently requires additional explanation. It only regenerates the current action. However, if previous actions have introduced "potential risk" that is non-trivial, the risk may not be mitigated through only one step of regeneration. A more detailed explanation may be needed.

- I am wondering how good the utility of the agent is after the framework is introduced. It seems that the framework is relatively conservative in choosing further actions, and the results seem to reflect this.

---

### Official Review · Reviewer_6FPE · 2025-11-03

**Soundness:** 3
**Presentation:** 3
**Contribution:** 3
**Rating:** 8
**Confidence:** 4

**Summary:**

This paper introduces RL-GUARD, a proactive safety framework for Large Language Model (LLM) agents.
To address the critical issue of cascading failures in multi-step tasks, RL-GUARD employs an actor-critic architecture to monitor an agent's trajectory, anticipate risks, trigger safety reflections, and select corrective actions.
A key innovation is a risk-conditioned reward model that provides fine-grained, step-level safety signals.
The authors also contribute a large-scale benchmark dataset with 45,598 human-annotated trajectories for training and evaluation.
Extensive experiments demonstrate that RL-GUARD significantly reduces risk while maintaining task utility, outperforming state-of-the-art baselines.

**Strengths:**

Well-Motivated Problem:​​ The research effectively highlights the critical issue of cascading failures in LLM agents, using a compelling example (Figure 1) to illustrate the real-world impact.

​​Sound Methodology:​​ The interplay between the critic (for risk forecasting) and the actor (for safe action selection) is logically sound. The pipeline is clearly described, and the risk-reflection mechanism is a practical solution.
​​
Comprehensive Experiments:​​ The paper provides extensive evaluations across three complementary benchmarks (RL-GUARD, ToolEmu, AgentHarm) and multiple LLMs (both proprietary and open-source), convincingly demonstrating the framework's effectiveness and generalizability.

​​High-Quality Resource:​​ The scale, diversity (8 risk categories, 40+ toolkits), and detail (step-level human annotations) of the released benchmark dataset constitute a major contribution that will facilitate future research. The provision of API-independent simulators further enhances reproducibility.
​​
Ablation Studies:​​ The ablations on key factors like the number of action candidates and the impact of reflection hints provide valuable insights into the framework's mechanics.

**Weaknesses:**

Clarity on Reward Model:​​ While the design intuition for the safety reward model is clear, a more detailed description of the reward model's training process(hypo paras, learning rate, loss curvs, etc.) is needed.​​

Computational Overhead:​​ While a 29% overhead (for GPT-4o) is a significant improvement over the SOTA baseline, it remains non-negligible for real-time applications. The paper could discuss potential optimizations, such as early termination strategies for the reflection loop.

​​Limited Discussion on Failure Cases:​​ The failure cases in Appendix A.5 are noted but not deeply analyzed. A more thorough discussion of the root causes (e.g., deceptive reasoning, unreliable information sources) and how the framework could be improved to handle them would strengthen the paper.

Given the rapid development of proactive agents, some recently emerged literatures are necessary to be compared or discussed in the context of related work, e.g., Proactive Agent: Shifting LLM Agents from Reactive Responses to Active Assistance, iclr'25, etc.

**Questions:**

Clarity on Reward Model:​​ While the design intuition for the safety reward model is clear, a more detailed description of the reward model's training process(hypo paras, learning rate, loss curvs, etc.) is needed.​​

Computational Overhead:​​ While a 29% overhead (for GPT-4o) is a significant improvement over the SOTA baseline, it remains non-negligible for real-time applications. The paper could discuss potential optimizations, such as early termination strategies for the reflection loop.

​​Limited Discussion on Failure Cases:​​ The failure cases in Appendix A.5 are noted but not deeply analyzed. A more thorough discussion of the root causes (e.g., deceptive reasoning, unreliable information sources) and how the framework could be improved to handle them would strengthen the paper.

---

### Note · Authors · 2026-01-11

I have read and agree with the venue's withdrawal policy on behalf of myself and my co-authors.